# About the Need for a More Adequate Way to Get an Understanding of the Experiencing of Aesthetic Items

**DOI:** 10.3390/bs13110907

**Published:** 2023-11-06

**Authors:** Claus-Christian Carbon

**Affiliations:** 1Department of General Psychology and Methodology, University of Bamberg, 96047 Bamberg, Bavaria, Germany; ccc@uni-bamberg.de; Tel.: +49-951-863-1860; 2Research Group EPÆG (Ergonomics, Psychological Aesthetics, Gestalt), 96047 Bamberg, Bavaria, Germany

**Keywords:** empirical aesthetics, art, experiencing art, aesthetics, ecological validity, dynamics, model, aesthetic theory

## Abstract

We live in times when neuroscientific methods have become standard methods that many researchers can easily use. While this offers excellent opportunities to understand brain activities linked with aesthetic processing, we face the problem of using sophisticated techniques without a proper and valid theoretical foundation of aesthetics. A further problem arises from sophisticated methods often demanding strict constraints in presenting and experiencing aesthetic stimuli. However, when experiencing aesthetic items, contextual factors matter, e.g., social and situational affordances are essential in triggering a true and deep “Kunsterlebnis” (Experience of Art). Additionally, in Art, it is often not the artwork as an object that matters but the close relationship *with* and the processing *of* the artwork. However, art is only one facet of the whole aesthetic domain, beside, e.g., design, architecture, everyday aesthetics, dance, literature, music, and opera. In the present paper, I propose a dynamic and holistic aesthetic perspective that includes the respective context, situation, cognitive and affective traits and state of the beholder, ongoing processes of understanding, Zeitgeist, and other cultural factors, which can be applied to different aesthetic domains. When ignoring such temporal and dynamic factors, we will not understand the qualia of aesthetic processing. These considerations might help researchers in the field of aesthetics to better understand the experiencing of aesthetic items of all kinds—if we ignore these factors, we are missing the essence of experiencing aesthetic items, especially artworks. We aim to sensitize and inform readers about these ideas to inspire a deeper understanding of experiencing aesthetic items and the advancement of a theoretical framework addressing the experiencing of aesthetics from different domains.

## 1. Introduction to Aesthetics

Aesthetics was originally a specialized branch of philosophy, examining the empirical and conceptual underpinnings of art, beauty, and taste. This area of inquiry has garnered significant academic interest for its role in elucidating the cognitive and experiential dimensions of human interactions with both natural and artificial aesthetic stimuli. From the beginning, there was confusion about whether aesthetics refer to ordinary objects or only to artworks [1]. There is also a lack of consistent definitions within this domain of research as to what aesthetic experience, aesthetic perception, or aesthetic evaluation and emotion exactly mean (for an elaborate discussion on this topic see [2]). Additionally, theorists of aesthetics developed very different ways of defining what aesthetics are and whether related phenomena refer to subjectivist or objectivist ways of assessing aesthetics [3]. Whereas the (pure) subjectivist view claims that all aesthetic experiences are based on personal processing, i.e., cognitive and affective appraisal and evaluation, with modulations possible through cultural and temporal factors, the (pure) objective view claims stable standards, eternal qualities and object-oriented properties that are universal across persons and cultures. While these are two distinct approaches to understanding aesthetic qualities, both lines become indistinct in practice. We often adopt a stance that combines elements from both perspectives. For example, they may maintain that there are certain, fundamental, universal standards of beauty (for instance, general preferences for such, see [4]), especially when only minimal resources are available [5], while also recognizing the significant influence of personal and cultural contexts, especially when the response format is not strongly constrained [6]. It is essential to understand that both objective and subjective elements contribute to our comprehension and recognition of art and beauty, so both should always be considered in combination or harmonized within an interactionist perspective [2].

Correspondingly large is the range of research questions regarding aesthetics, which are very heterogeneous and widespread. In the following, I will briefly introduce the fields that marked modern, scientific approaches to aesthetics, starting in the 19th century with empirical aesthetics and continuing with the recent developments of neuroaesthetics in the late 20th century.

Until the 18th century, philosophy lacked a clear concept of aesthetics. German philosopher Alexander Gottlieb Baumgarten introduced the term “aesthetics” in his PhD thesis *Meditationes philosophicae de nonnullis ad poema pertinentibus* [7] and further developed the concept of aesthetics in his ground-breaking work *Aesthetica* [8], which already utilized the Latin word for aesthetics in its title. Not until 1825 was the term introduced into English [9], although the interpretation of what aesthetics strictly refers to and whether it is more about sensory input or “lower cognition” (following Baumgarten’s original idea), about the interplay of understanding and imagination [10] or clear top-down or “higher cognition”, e.g., because it should be seen as an interpretative act [11], has been disputed ever since.

### 1.1. Empirical Aesthetics

With the advancement of scientific methods and a trend towards experimental research in the 19th century, interdisciplinary researchers such as Gustav Theodor Fechner (he was a professor of philosophy but, in fact, the first empirical psychologist of modern times, founding the essential methodological field of psychophysics) started to develop empirical methods to assess aesthetic phenomena of all kinds [12] and with a variety of methods, such as psychophysics [13]. Although empirical aesthetics can be considered as the initial topic of empirical psychological research due to Fechner’s studies in the middle of the 19^th^ century and despite its high relevance for human culture and being [14], empirical aesthetics was often identified as a side topic of psychology and was not developed much further until D. E. Berlyne’s introduction of *New Experimental Aesthetics* [15], which was characterized by a firm reliance on the physiological foundation of aesthetic phenomena.

### 1.2. Neuroaesthetics

The study of aesthetics is nowadays characterized by interdisciplinary and diverse research but very much influenced by Berlyne’s approach of including physiological and neurophysiological methods [16,17]. Recent developments in neuroscience and psychology [18] emerged as so-called “neuroaesthetics”, a term coined by Semir Zeki, a neurobiologist, in the very late days of the 20th century [19]. Zeki and his colleagues conducted some pioneering studies using neuroimaging techniques to examine the brain’s response to aesthetically relevant stimuli, such as paintings and photographs. Their research helped to understand how the brain responds to such aesthetically relevant stimuli, with a clear focus on investigating the role of the visual cortex in processing visual stimuli [20]. The aesthetic research subfield of neuroaesthetics is enriching our view on aesthetic phenomena by employing sophisticated techniques such as neuroimaging, including functional magnetic resonance imaging (fMRI), positron emission tomography (PET), and electroencephalography (EEG). With the help of these techniques, especially by combining them with more traditional experimental paradigms such as identification, evaluation and fast decision tasks, we can gain insights into where and when processes related to aesthetic experiences happen in the brain. Mostly, such highly methodologically sophisticated research comes at a price, typically leading to reductionist material and highly constrained test conditions and artificial environments where tests take place [21]. It is also important to note that most research in the field of empirical (neuro-)aesthetics does not utilise the great possibilities of modern time-sensitive methods such as reaction-timing, EEG, and MEG to obtain deeper insights into the processing and sub-processing and thus the temporal and process-oriented unfolding of an aesthetic experience [22].

## 2. Aesthetic Phenomena

Our present world is dominated by aesthetics—you will literally find aesthetic considerations everywhere. People take care of their outward appearance, consumers buy aesthetically pleasing food [23] and consumer products such as cars or smartphones [24], and homes, houses, and museums are trended towards appreciated aesthetics [25]. When looking at *Google Ngram* based on the *English-19* corpus [26], it is obvious that aesthetics plays a strong role, it even outperforms sports and topics around “Trump”, reaching nearly the same level of attention as politics in the retrieved literature. Beyond mere aesthetics, references to art are even more often found in the literature; see Figure 1.

There are a number of phenomena in aesthetics that seem to show, independent of culture, Zeitgeist, and the people who are involved in assessing aesthetics, that there is no thing such as the aesthetic object (e.g., an artwork or a design item) as such but that this emerges on the basis of our aesthetic experiences. This point will be briefly carved out by referring to several phenomena that illustrate this non-object-oriented approach. On the basis of these phenomena, we will present a conclusion that leads to preliminary ideas towards a theoretical framework of how we experience aesthetics.

### 2.1. Individual Taste

A first and very important finding is that there is no standard for good taste but that only at certain epochs, some groups, mostly elite groups, define what they consider to be good taste [27]. Actually, taste is a highly subjective matter. Honekopp [28] initially demonstrated by means of variance components that typical beholders of one culture share about half of the variance, with approximately 0.42 average interjudge agreement and 0.73 average retest reliability. These results, specifically for facial attractiveness, could be confirmed on similar levels for a series of domains, e.g., for liking artworks (private taste amounts to 75%) and the attractiveness of faces (private taste amounts to 40%) [29]—cf. [30].

### 2.2. Long-Term Change of Liking

Although our individual taste might be relatively stable, indicated by quite high retest reliabilities, as indicated before, we experience an ongoing change of what we like over time [31]. This is evident with fashion, where we rely on the fashion concepts of our current Zeitgeist. It is also evident that we cannot freely select among all possible designs due to a lack of offering possibilities—the market is, strictly speaking, limited to typical Zeitgeist trends, or at least it takes much more effort to find designs deviating from the current fashion trend, e.g., flare jeans in times when peg-top jeans are fashionable. Based on evolutionary universal preference theories, it is often assumed that round forms are preferred to more angular forms due to safety and pleasantness associations deeply rooted in our minds [4]. Such cycles of preferences for specific shapes dependent on Zeitgeist effects could be empirically revealed for the domain of car designs [26], where round forms dominated only in specific times, such as the 1950s and 2000s—see Figure 2.

In fact, especially for car designs, there was not only a unique trend for specific forms following such a cycle, but seemingly many car manufacturers showed very similar basic shapes, indicating that they all followed very general Zeitgeist trends (see Figure 2). It is, for instance, evident that angularly shaped car design dominated in the 1960s–1980s as illustrated in Figure 3.

Such Zeitgeist factors primarily work unconsciously but are very powerful. Form factors, content issues and style (even such seemingly easy qualities such as the depiction of perspective, see [33]) change over time and the history of art profoundly documents such dramatic changes [21]. There is no such thing as a perfect artwork, a perfectly beautiful church, or the most aesthetically pleasing landscape architecture. There can only be an approximation to something like that for a specific sample of people or individuals [34] within a certain era and within defined situations and situational and social demands [35]. Taste, acceptance, and enjoyment will change over time, and, as the changes are mostly smoothly transforming from one kind to another [36], they often go unnoticed. Only if we explicitly look back by looking up old picture books or historical advertisements, do we become aware of how strongly available aesthetics have changed, and thus our taste, too.

### 2.3. Short-Term Change of Liking

These Zeitgeist effects are often culturally modulated, and it seems that even big, independent organizations such as car companies synchronize with each other regarding the usage of specific Zeitgeist-dependent properties and qualities. However, also, on the individual level we can see drastic changes even over very short periods of time [37]. The idea that we are only bound to cultural context and social pressure [38] seems to be shortsighted. Actually, we undergo very fast transformations in liking and aesthetic evaluation due to adaptation effects. They happen quickly and sustainably [39] and have the power to substantially change mental representations and their evaluations, even if widely known, strongly represented, and reliably recognizable aesthetic items such as Leonardo’s Mona Lisa are used as target material [40]. When confronted with the alternative depiction of the Mona Lisa, the second version from Leonardo’s studio [41], now owned by the Prado Museum in Madrid, people typically react very negatively—they disapprove of this version, primarily due to its (original) colorfulness which is typical for Leonardo’s painting style and which the Louvre version is nowadays lacking due to its yellowed varnish. People are so accustomed to the yellowish-brownish color tones of the Louvre version with hardly any original color spectrum any longer visible that they reject all other versions, especially if they intensively use colors. This general and strong attitude can be modulated quickly and easily by letting participants elaborate the alternative, colorful Prado version. After having inspected the Prado version for just 10 min and reporting about different properties of this version, they showed an entirely different attitude pattern [40]: the more Prado-*esque* the version was (a morphing technique generated gradual intermediate versions between the Prado and the Louvre version), the higher the artistic quality. We revealed opposite effects when the Louvre version had to be initially considered; then, the participants evaluated the artistic quality more highly, the more Louvre-*esque* the version was (see Figure 4).

These findings are fascinating, as the Mona Lisa is typically quite stably represented. But still, a previously unfamiliar version takes over the role of the more convincing, artistic version. This indicates that familiarity is quickly achieved and that people need only a very short time to become accustomed to new material. However, they can then quickly adapt their attitudes and, probably, their taste. Indeed, similar effects were found regarding aesthetic appreciation, for instance, with employed material such as lip attractiveness after adaptation to lip fullness [42] or face attractiveness in photographs and art portraits [43].

### 2.4. Instant Change of Liking

There are long-term and short-term cycles of preferences, but we often experience sudden changes in liking due to extraordinary cognitive and emotional episodes. The reasons for such episodes are various, for instance, we are often delighted to recognize something we have desired or which we have not seen for a long time but feel strongly emotionally connected with. In the domain of aesthetics, we refer to such a concept or experience where an individual suddenly perceives or appreciates the aesthetic qualities of something in a profound or enlightening way simply as “Aesthetic Aha” [44]. This term blends the idea of having an “aha moment,” which is a sudden realization or insight (or as it is sometimes called, “from Oh to Ah”, see [45]) that pertains to the appreciation of liking and the sensory qualities of objects or experiences (see for an illustration Figure 5). This was especially shown for artworks, for instance, when people observed concrete items in barely decipherable Cubist paintings [46]. The impact of an “Aesthetic Aha” can be quite significant, both on an individual level and potentially on a broader societal or cultural level. It is thought that the increased liking you experience with an aesthetically relevant item due to Aesthetic Aha is the reward for making sense out of fuzzy or unclear patterns in order to motivate you to follow the way of sense-making [47].

### 2.5. Liking through Semantic Instability

It would be wrong to think of liking being changed in a linear and additive way on the basis of all the aforementioned effects towards a clear preference pattern. Rather, the complex temporal interplay of affirmative and negative attitudes towards aesthetic items, especially artworks, can often be described as an oscillating trajectory between understanding, acceptance, and liking and overtaxing, breaking of perceptual habits, and cognitive as well as emotional challenges. Muth and Carbon [48] call this phenomenon *semantic Instability* (*SeIns*—pronounced similar to “signs”). They have identified at least four different variants of *SeIns* [49] that all offer the opportunity of rewarding insight. This promise of late reward is often quite pleasurable due to “tentative prediction errors” [50]. As soon as we have reached a kind of seemingly stable status in semantics, e.g., because we understand parts of the aesthetic item or can assign it to a certain function, meaning, or association, we typically start into new domains of instability and so enable new promises of future reward that are so powerful that we are ongoingly involved in the processing of such an aesthetic item. For the domain of art, we often see a lifelong occupation with artworks that raise personal interest.

### 2.6. Conclusion: There Is No Such Thing as an Aesthetic Object as Such

All the facets of changing preferences with regard to an aesthetic item make clear that it is not about the physical qualities of such an entity but that only the psychological engagement with it is responsible for its assessment. Any kind of reliance on the mere physical nature of an entity will lead to misconceptions and create a naïve access to this entity. Still, the analysis of object properties is the most available approach to scientifically assess the experiencing of aesthetic items.

## 3. Some Thoughts about a Future Theoretical Framework of Experiencing Aesthetics

Undoubtedly, empirical aesthetics is now not only a promising emergent field of psychology and related scientific areas, but it is nowadays a sophisticated and respected research field, because its relevance is so apparent for many related fields such as engineering design, acoustic design, usability and ergonomics, software development, architecture, urban planning, or all artistic and culturally related domains: understanding why people like things, how artworks can deeply affect and touch people (when they experience the so-called “Kunsterlebnis”, i.e., a deep experience of art) and even transform people, why particular consumer products sell better than others, and, most importantly, why aesthetic items are often initially rejected due to their high degree of novelty, unfamiliarity, or innovativeness but are enjoyed or even loved exactly due to this quality after a period of accustomisation or mere usage. However, we still lack convincing universal aesthetic theories that solidly deal with the aforementioned phenomena because they mostly do not adequately address the object-independent (also called stimulus-oriented approach, see [2]) psychological phenomena around the dynamics of aesthetics and the strong and often highly idiosyncratic associations of aesthetic items with parts of individual learning histories. This does not neglect that object-inherent qualities such as peak-shift [51], prototypicality [52], novelty [53], and fluency [54] play a role in the assessment of aesthetics, but it would be wrong to propose that these effects are not modulating, nullifying, or even turning to the opposite as soon as contextual information [55], Zeitgeist factors [56], accustomisation [24], or associations [57] come into play. Essentially, we have to eliminate the idea that aesthetic items are aesthetically stable or even that there are general rules of aesthetics that always lead to high aesthetic qualities. Cultural changes and cultural peculiarities, as well as the whole field of art history, make very clear that people of different cultural backgrounds, stemming from different generations and having different educational or social statuses, show different, sometimes opposite attitudes towards specific aesthetic qualities. An excellent overview of conducting aesthetic research from different vantage points was made by Jacobsen [58] where he explicitly mentioned cultural, personal, and situational aspects and also referred to body-related [59] and mind-focused properties that all have an impact on the experiencing of aesthetics. Aesthetic evaluations can change over time, even within an individual, and can, even more fascinatingly, modulate within a very short period of time just by framing the aesthetic item differently. In fact, to integrate such factors in a future theoretical framework is essential to cover the exciting, dynamic qualities of aesthetic items, making them true psychologically relevant entities. Otherwise, we do not only miss some essential qualities of experiencing items, but misunderstand the fascination, awe, and transformative power of such items, particularly those from the domain of art.

Therefore, aesthetic processing should always be analyzed along a temporal dimension and with a perspective of dynamics due to ongoing cognitive and emotional processing when people are confronted with and interested in aesthetic items. In the specific case of analyzing aesthetic experience from the perspective of neuroaesthetics, this includes feedback and modulation loops that have the power to change the assessment and enjoyment of aesthetic items essentially. Such feedback loops are often lacking process models or are typically marginally referred to [22,60,61], but, like most processes in the brain [62], such feedback loops and recurrent structures overwriting the initial sensory signal are essential for the understanding of the experiencing of art.

A concrete step forward in this direction will always be to reflect the need to include some key dimensions and factors described above in planned (neuro-)aesthetic studies. This could be done by a tag list or flow diagrams addressing those variables. Such a prescriptive system will also assist us in detecting the limitations of any study to make more explicit and specifically clear which ecological factors might have been ignored or should be interpreted with caution. In this realm, we should always balance experimental control with ecological validity, as sketched out in Figure 6.

## 4. Author’s Bio

**Claus-Christian Carbon** (CCC) studied Psychology (Dipl.-Psych.), followed by Philosophy (M.A.), both at University of Trier, Germany. After receiving his PhD from the Freie Universität Berlin and his “Habilitation” at the University of Vienna, Austria, he worked at the Delft University of Technology, Netherlands and the University of Bamberg, Germany, where he currently holds a full professorship leading the Department of General Psychology and Methodology and the “Forschungsgruppe EPAEG”—a research group devoted to enhancing knowledge, methodology, and enthusiasm in the fields of cognitive ergonomics, psychological aesthetics, and Gestalt. He is the Dean of the Faculty of Human Sciences at the University of Bamberg. CCC is the author of more than 500 publications, including more than 200 peer-reviewed international journal articles, mainly addressing perceptual topics, and has conducted more than a dozen research projects with a total budget amount of approx. EUR 5 million on perception and marketing issues and is a renowned contributor and invited speaker at international research conferences. CCC is Editor-in-Chief of the scientific journal Art & Perception, Section Editor of Perception and i-Perception, Associate Editor of Frontiers in Psychology, Frontiers in Neuroscience, Journal of Perceptual Imaging, and Advances in Cognitive Psychology, and a member of the Editorial Boards of Open Psychology and Leadership, Education, and Personality. Since 2023, CCC has been an elected, ordinary member of the European Academy of Sciences and Arts.

## Figures and Tables

**Figure 1 behavsci-13-00907-f001:**
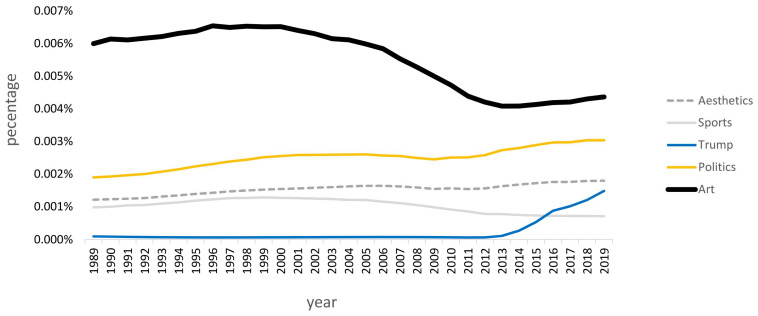
Google Ngram for the search words “aesthetics”, “sports”, “Trump”, “politics”, and “art”, based on the English-19 corpus. Data were retrieved on 18 February 2023 by R library ngram’s function ngram. Exact numbers may vary from data request to data request.

**Figure 2 behavsci-13-00907-f002:**
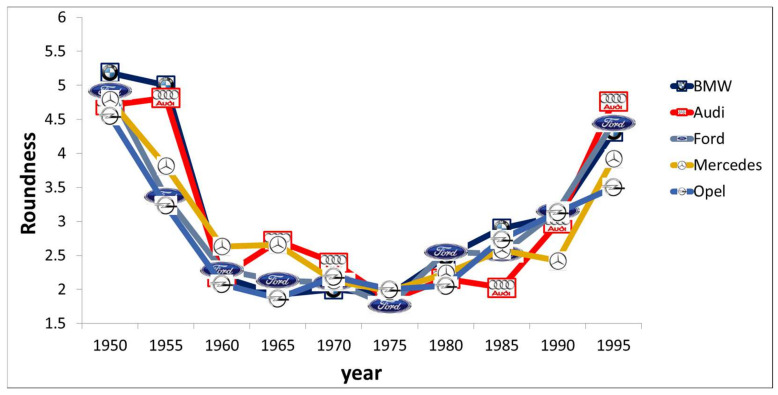
The German car market from 1950–2000 on the basis of the car brands BMW, Audi, Ford (a US brand, but the European design variants are mainly issued in Germany), Mercedes-Benz, and Opel (formerly also a US brand but which was always designed in Germany). Data from Carbon [32].

**Figure 3 behavsci-13-00907-f003:**
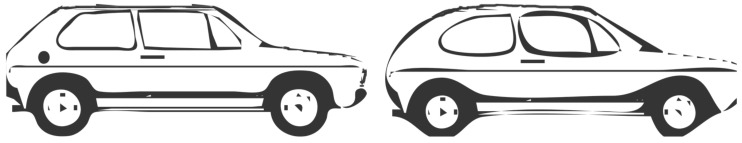
Car design on the basis of the VW Golf I (“Rabbit”) in an angular (**left**) vs. curved (**right**) design. Own depictions made by CCC.

**Figure 4 behavsci-13-00907-f004:**
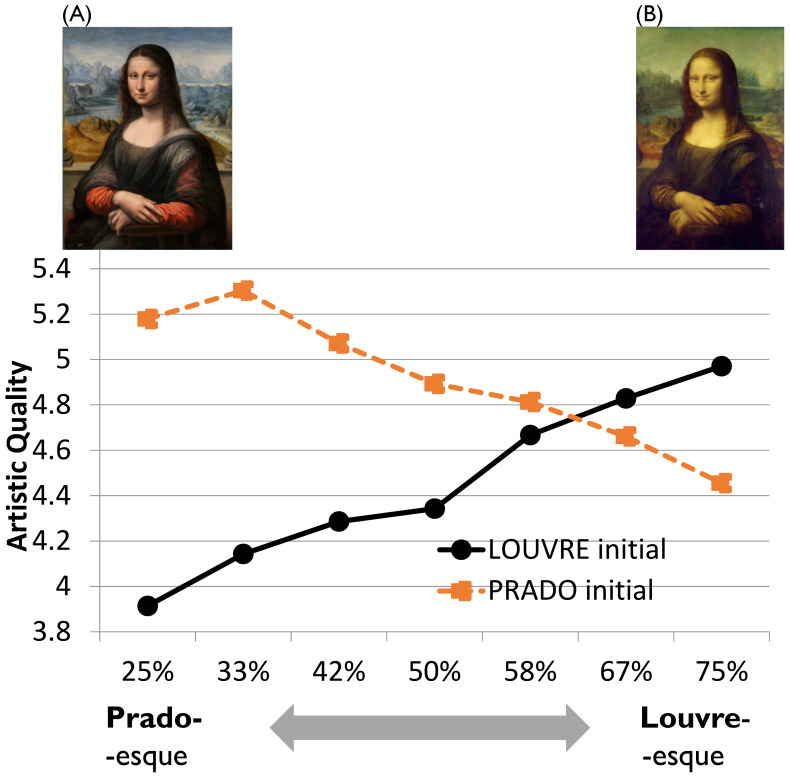
Two times the Mona Lisa ((**A**): the Prado version; (**B**): the Louvre version). Participants were exposed either to the Prado (**A**) or the Louvre (**B**) version; although participants typically reject the more Prado-esque version of the Mona Lisa (realized by morphing between both versions), they actually preferred the Prado-seque versions (in terms of artistic quality) when they first inspected and considered the Prado version as seen in the dissociated data pattern.

**Figure 5 behavsci-13-00907-f005:**
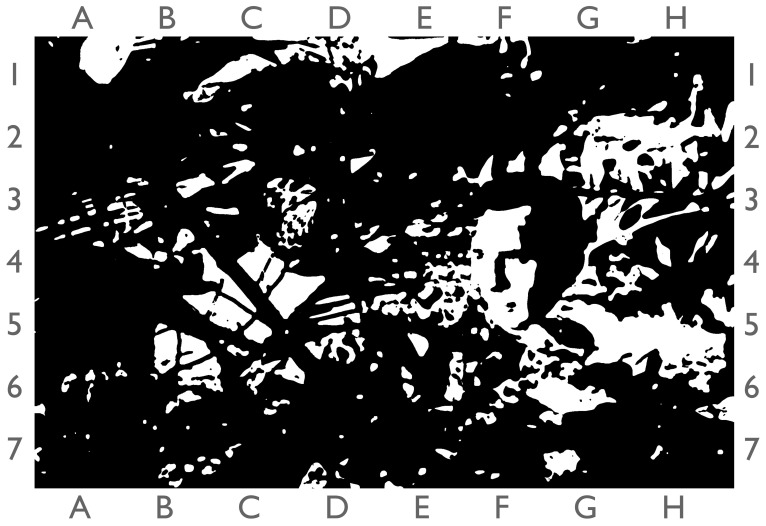
Demonstration of the Aesthetic Aha by means of so-called “Mooneyized” (blackened and whitened) picture of a tropical landscape where the face of Jack Kerouac (American writer and poet, 1922–1969) is overlayed (sector F4/G4). As soon as you recognize the face or even Kerouac himself, your liking of the whole scene is heightened. © CCC 2023.

**Figure 6 behavsci-13-00907-f006:**
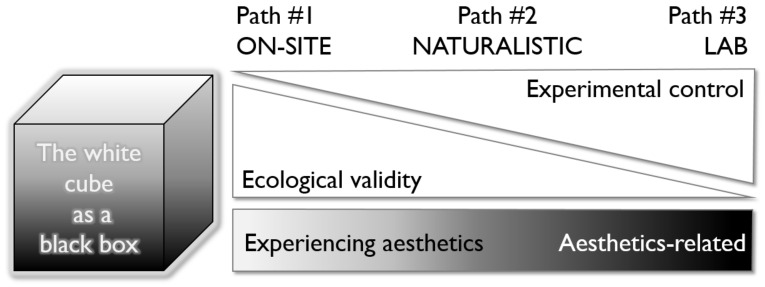
The white cube as a black box and how to conduct studies in the field of experiencing aesthetics, based on the considerations of Carbon [14]. © CCC 2023.

## Data Availability

No data available for this article.

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
