# Peer review of "About the Need for a More Adequate Way to Get an Understanding of the Experiencing of Aesthetic Items"

_behavsci, 2023, doi:10.3390/bs13110907_

Round 1

Reviewer 1 Report

Comments and Suggestions for Authors

The present manuscript primarily provides a short synoptic overview of the author’s research on the stability of aesthetic judgment. I enjoyed reading it as a review or summary; however, based on the title, the abstract, and the introduction, I expected a different type of paper—a guiding theoretical framework (this term is used by the author) for future research, which takes into account an important aspect that has been overlooked until now. This is also one of my main concerns with the current submission. From my perspective, it falls short of the expectations set in the introduction. Either the paper needs to be rewritten to align with the raised expectations or the expectations need to be adjusted to match the current content. It's worth noting that over most sections, the manuscript summarizes already published research. The actual 'solution,' the theoretical framework, is presented in section 3 and comprises no more than 14 lines of text.

Secondly, the manuscript is not integrated into the ongoing debate within the neuroaesthetics/empirical aesthetics community. Many aspects discussed in the manuscript, such as the debate on object-inherent and object-independent qualities or the confusion around whether aesthetics refers solely to artworks or also to "ordinary objects," have been extensively addressed in recent publications but go unnoticed in the present manuscript.

Third and related to the last example, I observe conceptual instability in the use of terms such as "aesthetic item" throughout the paper, while also recurrently stating "there is no aesthetic object as such." What is the difference between "item" and "object"?

The latter issue might be connected to a general lack of elaboration, which is my fourth concern. Many terms, concepts, and theoretical positions are presented in an overly abbreviated manner. For instance, the historical overview is, in my opinion, oversimplified and lacks sufficient differentiation. Moreover, while some aspects, such as "subjectivist or objectivist approaches to assessing aesthetics", may be understood by readers familiar with the general topic, this level of expertise cannot be assumed for the readership of the target journal. Relatedly, some terms, for instance, 'Kunsterlebnis,' appear only once in the abstract.

Fifth, as a minor point, the manuscript in its current form contains several typos and syntactic errors (e.g., "spoirt," "Prado-esque," "Hürtel").

Comments on the Quality of English Language

see fifths point above

Author Response

Reviewer 1

Comments and Suggestions for Authors

The present manuscript primarily provides a short synoptic overview of the author’s research on the stability of aesthetic judgment. I enjoyed reading it as a review or summary; however, based on the title, the abstract, and the introduction, I expected a different type of paper—a guiding theoretical framework (this term is used by the author) for future research, which takes into account an important aspect that has been overlooked until now. This is also one of my main concerns with the current submission. From my perspective, it falls short of the expectations set in the introduction. Either the paper needs to be rewritten to align with the raised expectations or the expectations need to be adjusted to match the current content. It's worth noting that over most sections, the manuscript summarizes already published research. The actual 'solution,' the theoretical framework, is presented in section 3 and comprises no more than 14 lines of text.

** RESPONSE: Thanks for your review; I share your view and have tried to lower the expectations on the one hand by changing the title, the abstract, the title of chapter 3 (now “Some thoughts about a future theoretical framework of experiencing aesthetics”), but also to extend the chapter on developing a future theoretical framework.

The title is now “About the need for a more adequate way to get an understanding of the experiencing of aesthetic items“. In the abstract, I added a sentence at the end that we are far from being able to providing a theoretical framework yet: “These considerations might help researchers in the field of aesthetics to understand better the experiencing of aesthetic items.” The last chapter now contains additional parts. It reads now:

“Undoubtedly, empirical aesthetics is now not only a promising emergent field of psychology and related scientific areas, but it is nowadays a sophisticated and respected research field, because its relevance is so apparent for many related fields such as engineering design, acoustic design, usability and ergomomics, software development or architecture, urban planning or all art- and culturally related domains: understanding why people like things, how artworks can deeply affect and touch people (when they experience the so-called “Kunsterlebnis”, i.e. a deep experience of art) and even transform people, why particular consumer products are better sold than others, and most importantly why aesthetic items are often rejected at first stance due to their high degree of novelty, unfamiliarity or innovativeness but are enjoyed or even loved exactly due to this quality after a while of elaboration or mere usage. However, we still lack convincing universal aesthetic theories that solidly deal with the aforementioned phenomena because they mostly do not adequately address the object-independent [also called stimulus-oriented approach, see 2] psychological phenomena around the dynamics of aesthetics and the strong and often highly idiosyncratic associations of aesthetic items with parts of individual learning histories. This does not neglect that object-inherent qualities such as peak-shift [51], prototypicality [52], novelty [53] and fluency [54] play a role in the assessment of aesthetics, but it would be wrong to propose that these effects are not modulating, nullifying or even turning to the opposite as soon as context information [55], Zeitgeist factors [56], elaboration [24] or associations [57] come into play. Essentially, we have to eliminate the idea that aesthetic items are aesthetically stable or even that there are general rules of aesthetics that always lead to high aesthetic qualities. Cultural changes and cultural peculiarities, as well as the whole field of art history, make very clear that people of different cultural backgrounds, stemming from different generations and having different educational or social statuses, show different, sometimes opposite attitudes towards specific aesthetic qualities. An excellent overview of conducting aesthetic research from different vantage points was made by Jacobsen [58] where he explicitly mentioned cultural, personal and situational aspects and also referred to body-related [59] and mind-focused properties that all have an impact on the experiencing of aesthetics. Aesthetic evaluations can change over time, even within an individual, and can, even more fascinating, modulate within a very short period of time just by framing the aesthetic item differently. In fact, to integrate such factors in a future theoretical framework is essential to cover the exciting, dynamic qualities of aesthetic items making them true psychologically relevant entities.

Therefore, aesthetic processing should always be analyzed along a temporal dimension and with a perspective of dynamics due to ongoing cognitive and emotional processing when people are confronted with and interested in aesthetic items. In the specific case of analyzing aesthetic experiencing from the perspective of neuroaesthetics, this includes feedback and modulation loops that have the power to essentially change the assessment and enjoyment of aesthetic items. Such feedback loops often are missing in process models or are typically marginally referred to [22,60,61], but like most processes in the brain [62], such feedback loops and recurrent structures overwriting the initial sensory signal are essential for the understanding of the experiencing of art.”

**

Secondly, the manuscript is not integrated into the ongoing debate within the neuroaesthetics/empirical aesthetics community. Many aspects discussed in the manuscript, such as the debate on object-inherent and object-independent qualities or the confusion around whether aesthetics refers solely to artworks or also to "ordinary objects," have been extensively addressed in recent publications but go unnoticed in the present manuscript.

** RESPONSE: Thanks for this important feedback. The presented paper is a reflection of the keynote talk I gave at the neuroaesthetics symposium in Suceava at the beginning of this year where I mainly referred to own examples illustrating my line of argument. However, it is true that many others have raised similar issues, so I have included such sources in the text as well and this aligns nicely with the major arguments I provide through the paper.

I have added the following sources (I am also greatful for sources the reviewers drew my attention on which are also included on the following list):

Aleem, H.; Grzywacz, N.M. The Temporal Instability of Aesthetic Preferences. Psychol. Aesthet. Creat. Arts. 2023, doi:10.1037/aca0000543.

Aleem, H.; Grzywacz, N.M. The Temporal Instability of Aesthetic Preferences. Psychol. Aesthet. Creat. Arts. 2023, doi:10.1037/aca0000543.

Bar, M.; Neta, M. Humans prefer curved visual objects. Psychol Sci 2006, 17, 645-648.

Bar, M.; Neta, M.; Linz, H. Very first impressions. Emotion 2006, 6, 269-278.

Carbon, C.C. Ecological art experience: How we can gain experimental control while preserving ecologically valid settings and contexts. Frontiers in Psychology 2020, 11, 1-14.

Chatterjee, A. Prospects for a cognitive neuroscience of visual aesthetics. Bulletin of Psychology and Arts 2004, 4, 55-60.

Jacobsen, T. Bridging the arts and sciences: A framework for the psychology of aesthetics. Leonardo 2006, 39, 155-162.

Kaplan, E.; Mukherjee, P.; Shapley, R.M. Information filtering in the lateral geniculate nucleus. In Contrast sensitivity, Shapley, R., Lam, D.M.-K., Eds.; MIT Press: Cambridge, MA, 1993.

Kubovy, M. Neuroaesthetics: Maladies and Remedies. Art & Perception 2020, 8, 1-26, doi:10.1163/22134913-20191138.

Kubovy, M. The psychology of perspective and Renaissance art; Cambridge University Press: Cambridge, UK, 1986.

Kühnapfel, C.; Fingerhut, J.; Brinkmann, H.; Ganster, V.; Tanaka, T.; Specker, E.; Mikuni, J.; Güldenpfennig, F.; Gartus, A.; Rosenberg, R.; et al. How do we move in front of art? How does this relate to art experience? Linking movement, eye tracking, emotion, and evaluations in an ecologically-valid gallery setting; 2022.

Leder, H.; Belke, B.; Oeberst, A.; Augustin, D. A model of aesthetic appreciation and aesthetic judgments. Br. J. Psychol. 2004, 95, 489-508.

Magnusson, C.; Anastassova, M.; Paneels, S.; Rassmus-Gröhn, K.; Rydeman, B.; Randall, G.; Ortiz Fernandez, L.; Bouilland, S.; Pager, J.; Hedvall, P.O. Stroke and Universal Design. Transforming Our World through Design, Diversity and Education 2018, 256, 854-861, doi:10.3233/978-1-61499-923-2-854.

Mojet, J.; Köster, E. The Dynamics of Liking. In Time‐Dependent Measures of Perception in Sensory Evaluation; 2017; pp. 124-155.

Ortlieb, S.A.; Fischer, U.C.; Carbon, C.C. Enquiry into the origin of our ideas of the sublime and beautiful: Is there a male gaze in empirical aesthetics? Art & Perception 2016, 4, 205-224.

Pelowski, M.; Markey, P.S.; Lauring, J.O.; Leder, H. Visualizing the impact of art: An update and comparison of current psychological models of art experience. Frontiers in Human Neuroscience 2016, 10, doi:10.3389/fnhum.2016.00160.

Storie, M.; Vining, J. From Oh to Aha: Characteristics and Types of Environmental Epiphany Experiences. Hum Ecol Rev 2018, 24, 155-179.

Szubielska, M.; Imbir, K.; Fudali-Czyz, A.; Augustynowicz, P. How Does Knowledge About an Artist's Disability Change the Aesthetic Experience? Advances in Cognitive Psychology 2020, 16, 150-159, doi:10.5709/acp-0292-z.

Uhde, A.; Hassenzahl, M. Towards a Better Understanding of Social Acceptability. Extended Abstracts of the 2021 Chi Conference on Human Factors in Computing Systems (Chi'21) 2021, doi:10.1145/3411763.3451649.

Wassiliwizky, E.; Menninghaus, W. Why and How Should Cognitive Science Care about Aesthetics? Trends Cogn Sci 2021, 25, doi:10.1016/j.tics.2021.03.008.

Third and related to the last example, I observe conceptual instability in the use of terms such as "aesthetic item" throughout the paper, while also recurrently stating "there is no aesthetic object as such." What is the difference between "item" and "object"?

** RESPONSE: I attack the concept of an object being investigated by object properties to get insights into the aesthetic processing of such an entity, but in the end, we have to talk about the object, whether it is an artwork, a consumer product or a kitschy souvenir. So in this text I do not differentiate between object and item (and entity) but we should not focus on the properties of such an object (or item/entity) to get a deeper understanding of the quality of experiencing it. **

The latter issue might be connected to a general lack of elaboration, which is my fourth concern. Many terms, concepts, and theoretical positions are presented in an overly abbreviated manner. For instance, the historical overview is, in my opinion, oversimplified and lacks sufficient differentiation. Moreover, while some aspects, such as "subjectivist or objectivist approaches to assessing aesthetics", may be understood by readers familiar with the general topic, this level of expertise cannot be assumed for the readership of the target journal. Relatedly, some terms, for instance, 'Kunsterlebnis,' appear only once in the abstract.

** RESPONSE: I agree that some topics are only discussed shortly but the aimed perspectives format is typically not that going deeply as other formats where very long and elaborated pieces are explicitly looked for. I have taken care to elaborate a bit on the subjectivist vs. objectivist approaches to also catch up with readers not so familiar with these concepts. This section read now: “From the beginning, there was confusion about whether aesthetics refer to ordinary objects or only to artworks [1]. There is also a lack of consistent definitions within this domain of research what aesthetic experience, aesthetic perception or aesthetic evaluation and emotion exactly means [see for an elaborate discussion on this topic 2]. Additionally, theorists of aesthetics developed very different ways of defining what aesthetics are and whether related phenomena refer to subjectivist or objectivist ways of assessing aesthetics [3]. Whereas the (pure) subjectivist view claims that all aesthetic experiences are based on personal processing, so cognitive and affective appraisal and evaluation, with modulations possible through cultural and temporal factors, the (pure) objective view claims stable standards, eternal qualities and object-oriented properties that are universal across persons and cultures. While these are two distinct approaches to understanding aesthetic qualities, both lines become indistinct in practice. We often adopt a stance that combines elements from both perspectives. For example, they may maintain that there are certain, fundamental, universal standards of beauty [for instance, general preferences for, see 4], especially when only minimal resources are available [5], while also recognizing the significant influence of personal and cultural contexts, especially when the response format are not strongly constrained [6]. It is essential to understand that both objective and subjective elements contribute to our comprehension and recognition of art and beauty, so both should always be considered in combination or harmonized within an interactionist perspective [2].

About “Kunsterlebnis”: that is right and I have now included it also in the body text where it fits quite well with the there elaborated ideas on experiencing art. **

Fifth, as a minor point, the manuscript in its current form contains several typos and syntactic errors (e.g., "spoirt," "Prado-esque," "Hürtel").

** RESPONSE:

  • Sports: addressed
  • Prado-esque: this is on purpose, it aims to describe pictures that go towards the Prado-version; if there are better ways to describe this, I am very happy to hear about them
  • Härtel: addressed – thanks for your focused eye on these subtle but important typos. **

Reviewer 2 Report

Comments and Suggestions for Authors

This manuscript discusses the need for a new approach in the research on aesthetics. Even though I believe this is an interesting and quite important topic, it is not new, and there are several limitations to the manuscript.

First, I am having difficulties understanding the main aim of the paper. The title suggests a discussion of the need for a new approach. The abstract promises the proposal of a “dynamic and holistic aesthetic perspective that includes” (line 17), inter alia, the context, situation, and cultural factors. The main text offers an introduction to the (Neuro)aesthetics and an overview of aspects of liking, rather than focusing on aesthetic experiences or processing overall. The conclusion depicts mainly a differentiation between potential individual and object properties. Elaborating on what are the old approaches and what is/are (a) potential new approach/es while focusing specifically on the aspects mentioned in the abstract (i.e., “respective context, situation, cognitive and affective traits and state of the beholder, ongoing processes of understanding, Zeitgeist and other cultural factors”; lines 18–19) would help the reader understand the structure and main aim of the paper.

The abstract does not mention different (aesthetic) domains but only talks about the (aesthetic) experience of art. Would it not make sense if the abstract mentioned others besides art, since different domains are addressed in the main text? And what is the relevance of using the German term "Kunsterlebnis" (line 15)?

Has the approach suggested by the author not already been discussed or proposed in research (e.g., Jacobsen, 2006; Wassiliwizky & Menninghaus, 2021; for a review of some models of art experience, see Pelowski et al., 2016)? What exactly is new about this approach?

The conclusions and the last section fall too short. I would suggest the author to summarize, for instance, the new approach, what is new about this approach, and how exactly it contributes to the field of Neuroaesthetics. In particular, the clear reference to the so-called “Neuroaesthetics” is low. Mentioning it in the abstract and outlining the historical background in the introduction, the reader expects a stronger thematization of this subdiscipline than ultimately takes place.

I would also like to give some stylistic notes.

First, with respect to Figure 1, the author states (beginning in line 80) that aesthetics “outperforms sports and topics around “Trump”.” Using the same search words as well as applying the same corpus, the Figure I receive reveals a different picture: “Aesthetics” never outperformed sports, and “Trump” only a small amount until the year 2013. Could it be that there were other settings that were not mentioned in the paper?

Second, there is a small spelling error in the Figure 1 note. I assume that instead of “spoirts,” it should read “sports.”  

Third, could it be that the sentences in lines 7275 and 155159 are incomplete?

References

Jacobsen, T. (2006). Bridging the arts and sciences: A framework for the psychology of aesthetics. Leonardo, 39(2), 155–162. https://doi.org/10.1162/leon.2006.39.2.155

Pelowski, M., Markey, P. S., Lauring, J. O., & Leder, H. (2016). Visualizing the impact of art: An update and comparison of current psychological models of art experience. Frontiers in Human Neuroscience, 10, 160. https://doi.org/10.3389/fnhum.2016.00160 

Wassiliwizky, E., & Menninghaus, W. (2021). Why and how should cognitive science care about aesthetics? Trends in Cognitive Sciences, 25(6), 437–449. https://doi.org/10.1016/j.tics.2021.03.008

Author Response

Reviewer 2

This manuscript discusses the need for a new approach in the research on aesthetics. Even though I believe this is an interesting and quite important topic, it is not new, and there are several limitations to the manuscript.

** RESPONSE: Thanks you for your time and effort that you have put into your evaluation and your suggestions for improving this manuscript. I will take the opportunity to respond to all of your raised points in the following. **

First, I am having difficulties understanding the main aim of the paper. The title suggests a discussion of the need for a new approach. The abstract promises the proposal of a “dynamic and holistic aesthetic perspective that includes” (line 17), inter alia, the context, situation, and cultural factors. The main text offers an introduction to the (Neuro)aesthetics and an overview of aspects of liking, rather than focusing on aesthetic experiences or processing overall. The conclusion depicts mainly a differentiation between potential individual and object properties. Elaborating on what are the old approaches and what is/are (a) potential new approach/es while focusing specifically on the aspects mentioned in the abstract (i.e., “respective context, situation, cognitive and affective traits and state of the beholder, ongoing processes of understanding, Zeitgeist and other cultural factors”; lines 18–19) would help the reader understand the structure and main aim of the paper.

** RESPONSE: I made the aim of the paper explicit now, at the end of the abstract: “These considerations might help researchers in the field of aesthetics to understand better the experiencing of aesthetic items of all kinds. We aim to sensitize the readers about these ideas to inspire the advancement of a theoretical framework addressing the experiencing of aesthetic items from different domains.”

I have also added a statement on the lack of processing-oriented research despite usage of highly time-sensitive measures like reaction timing, EEG and MEG at the end of chapter 1: “It is also important to note that most research in the field of empirical (neuro-)aesthetics does not use the great possibilities of modern time-sensitive methods like reaction-timing, EEG and MEG to get deeper insights into the processing and sub-processing and so the temporal and process-oriented unfolding of an aesthetic experience [see 22]”.

**

The abstract does not mention different (aesthetic) domains but only talks about the (aesthetic) experience of art. Would it not make sense if the abstract mentioned others besides art, since different domains are addressed in the main text? And what is the relevance of using the German term "Kunsterlebnis" (line 15)?

** RESPONSE: Thanks for the suggestion of adding other domains; as this was exactly the aim of the paper to be not just concerned about the art domain, this is a very important point. I have added the explicit notion of other notions now already in the abstract:

However, art is only one facet of the whole aesthetic domain, besides, e.g., design, architecture, everyday aesthetics, dance, literature, music, and opera. In the present paper, I propose a dynamic and holistic aesthetic perspective that includes the respective context, situation, cognitive and affective traits and state of the beholder, ongoing processes of understanding, Zeitgeist and other cultural factors, which can be applied to different aesthetic domains. When ignoring such temporal and dynamic factors, we will not understand the qualia of aesthetic processing. These considerations might help researchers in the field of aesthetics to understand better the experiencing of aesthetic items of all kinds. We aim to sensitize the readers about these ideas to inspire the advancement of a theoretical framework addressing the experiencing of aesthetic items from different domains.

About “Kunsterlebnis”: I pick up the term now within the body text, too, so that this technical term is embedded in the whole considerations conveyed by the text.

**

Has the approach suggested by the author not already been discussed or proposed in research (e.g., Jacobsen, 2006; Wassiliwizky & Menninghaus, 2021; for a review of some models of art experience, see Pelowski et al., 2016)? What exactly is new about this approach?

** RESPONSE: The manuscript is based on a keynote held by the author, so contains a series of sources from the author, but you are right, these important sources which also worked as inspirations for developing further ideas were not referred to up to now. This has now been changed. I particularly benefited from the model of Jacobsen (2006) which is now included and discussed within the paper’s chapter 3. I also refer to Pelowski et al’s (2016) and Leder et al.’s (2004) papers in the last part of chapter 3. I also added Chatterjee (2004).

The conclusions and the last section fall too short. I would suggest the author to summarize, for instance, the new approach, what is new about this approach, and how exactly it contributes to the field of Neuroaesthetics. In particular, the clear reference to the so-called “Neuroaesthetics” is low. Mentioning it in the abstract and outlining the historical background in the introduction, the reader expects a stronger thematization of this subdiscipline than ultimately takes place.

** RESPONSE: I have extended the last section which indeed was quite short. I have also added some specific notions about neuroaesthetics. **

I would also like to give some stylistic notes.

** RESPONSE: Thanks for putting also effort in these details which helped us to improve the ms. **

First, with respect to Figure 1, the author states (beginning in line 80) that aesthetics “outperforms sports and topics around “Trump”.” Using the same search words as well as applying the same corpus, the Figure I receive reveals a different picture: “Aesthetics” never outperformed sports, and “Trump” only a small amount until the year 2013. Could it be that there were other settings that were not mentioned in the paper?

** RESPONSE: I have checked Figure 1 and the involved corpora. I have used the R library “ngramr” which produces nearly the same results BUT when I checked and re-checked the results always varied a bit (also for the ngram internet interface freely available via https://books.google.com/ngrams) —see an example that I’ve produced today.

This is quite odd and seems to indicate that the whole algorithm is not reliable. I took the following action: 1) I have added the statement “Exact numbers may vary from data request to data request.” to the figure’s caption and 2) I have added more information about how I retrieved the data via ngram R function. **

Second, there is a small spelling error in the Figure 1 note. I assume that instead of “spoirts,” it should read “sports.” 

** RESPONSE: Thanks for spotting this; resolved. **

Third, could it be that the sentences in lines 72–75 and 155–159 are incomplete?

** RESPONSE: Thanks! This was addressed; it now reads: “With the help of these techniques, especially by combining them with more traditional experimental paradigms like identification, evaluation and fast decision tasks, we can gain insights into where and when processes related to aesthetic experiences happen in the brain.” **

References

Jacobsen, T. (2006). Bridging the arts and sciences: A framework for the psychology of aesthetics. Leonardo, 39(2), 155–162. https://doi.org/10.1162/leon.2006.39.2.155

Pelowski, M., Markey, P. S., Lauring, J. O., & Leder, H. (2016). Visualizing the impact of art: An update and comparison of current psychological models of art experience. Frontiers in Human Neuroscience, 10, 160. https://doi.org/10.3389/fnhum.2016.00160

Wassiliwizky, E., & Menninghaus, W. (2021). Why and how should cognitive science care about aesthetics? Trends in Cognitive Sciences, 25(6), 437–449. https://doi.org/10.1016/j.tics.2021.03.008

** RESPONSE: Thanks for the concrete sources, they were all very helpful and were included in the revision of the manuscript. **

Reviewer 3 Report

Comments and Suggestions for Authors

The article takes an interesting perspective on aesthetics and attempts to find a more solid ground for aesthetic research. There is no doubt that it pins down some crucial developments that help us analyse aesthetic phenomena that cannot be approached easily.

However, the argumentation why in the process of evaluation the Prado-esque Mona Lisa becomes more valuable is not entirely clear. Do we lack any ground for drawing conclusions? Is it somehow dependent on other factors that are not necessarily aesthetic? How does the psychological factor impact on the shift in evaluation? The (end-)suggestion that the outcome of aesthetic evaluation is heavily dependent on "cognitive and emotional processing" is definitely to the point but could possibly be developed and specified a little bit further.

Author Response

Reviewer 3

Comments and Suggestions for Authors

The article takes an interesting perspective on aesthetics and attempts to find a more solid ground for aesthetic research. There is no doubt that it pins down some crucial developments that help us analyse aesthetic phenomena that cannot be approached easily.

** RESPONSE: Thanks for your positive overall feedback! **

However, the argumentation why in the process of evaluation the Prado-esque Mona Lisa becomes more valuable is not entirely clear. Do we lack any ground for drawing conclusions? Is it somehow dependent on other factors that are not necessarily aesthetic? How does the psychological factor impact on the shift in evaluation? The (end-)suggestion that the outcome of aesthetic evaluation is heavily dependent on "cognitive and emotional processing" is definitely to the point but could possibly be developed and specified a little bit further.

** RESPONSE: I have taken care to be more specific in this regard. It reads now: “These findings are fascinating as the Mona Lisa is typically quite stably represented. But still, a previously unfamiliar version takes over the role of the more convincing, artistic version. This indicates that familiarity is quickly achieved and that people need only a very short time to elaborate on new material. However, then they can fast adapt their attitudes and probably taste. Indeed, similar effects were found regarding aesthetic appreciation, for instance, with employed material like lip attrac-tiveness after adaptation to lip fullness [41] or face attractiveness in photographs and art portraits [42].”. **

Round 2

Reviewer 2 Report

Comments and Suggestions for Authors

I thank the author for making such detailed revisions to the original version of the manuscript.

Although most of my comments have been addressed, the main concern remains: the originality of the work. I am still having difficulties understanding its contribution to the scholarship, as it mainly summarizes and partly discusses an existing research debate. Therefore, it represents more of a reminder of the relevance of this issue. So, what exactly is new about the proposed perspective (lines 17–18)? Explicitly stating its novelties in the manuscript seems important for the justification of the present research.

Based on the author's detailed revision, I have one more comment. The author states the aim is to “sensitize the readers about these ideas to inspire the advancement of a theoretical framework addressing the experiencing of aesthetic items from different domains” (lines 24–25). From the point of view of the author, do the readers need to be sensitized or do researchers need to find “a more adequate way to get an understanding of the experiencing of aesthetic items” (lines 1–2)? In addition, since this topic is not new, it would be worthwhile to discuss how to implement this new, appropriate path.

Author Response

Response to Reviewer 2 (round 2)

I thank the author for making such detailed revisions to the original version of the manuscript.

*** RESPONSE: Thanks for this and your time you’ve already invested in this review process to improve the current version of the manuscript! ***

Although most of my comments have been addressed, the main concern remains: the originality of the work. I am still having difficulties understanding its contribution to the scholarship, as it mainly summarizes and partly discusses an existing research debate. Therefore, it represents more of a reminder of the relevance of this issue. So, what exactly is new about the proposed perspective (lines 17–18)? Explicitly stating its novelties in the manuscript seems important for the justification of the present research.

*** RESPONSE: I am happy that most concerns have been resolved now. Like most new approaches, also this scientific work is based on pre-works of others and my own group. And indeed, it is a compilation of different approaches which are all promising in guiding researchers to the right direction to unveil and understand the experiencing of aesthetic items better or more adequately. What is really new is that it makes clear that not only factors like context, situation, cognitive and affective traits and state of the beholder, ongoing processes of under-standing, Zeitgeist and other cultural factors are influencing factors for aesthetic experiencing, but that all attempts to ignore them are misleading and will focus on qualities which are not any more covered by true experiencing. Ina  way, this manuscript is much more radical in stating that such research is ill-defined and we have to make the move towards more ecological validity as we lose the real phenomenon on the way and so can never grasp the experiencing underlying aesthetic processing.

To make this clearer, I have added to the abstract (line 24): “—if we ignore these factors we are missing the essence of experiencing aesthetic items, especially artworks”. Furthermore, I added the following statement to the discussion (lines 294ff.): “Otherwise, we do not only miss some essential qualities of experiencing items, but misunderstand the fascination, awe and transformative power of such items, particularly those from the domain of art.“ ***

Based on the author's detailed revision, I have one more comment. The author states the aim is to “sensitize the readers about these ideas to inspire the advancement of a theoretical framework addressing the experiencing of aesthetic items from different domains” (lines 24–25). From the point of view of the author, do the readers need to be sensitized or do researchers need to find “a more adequate way to get an understanding of the experiencing of aesthetic items” (lines 1–2)? In addition, since this topic is not new, it would be worthwhile to discuss how to implement this new, appropriate path.

*** RESPONSE: I agree, sensitizing the readership is not enough, it is really about sensitizing AND deeper understanding for these aesthetic phenomena which I refer to within the paper. Therefore, I have adapted this part of the abstract: “We aim to sensitize and inform the readers about these ideas to inspire deeper understanding of experiencing aesthetic items and the advancement of a theoretical framework addressing the experiencing of aesthetics from different domains.“

As it is essential to implement this path of research, I have added a short description to the discussion how this could happen: “A concrete step forward in this direction will always be to reflect the need to include some key dimensions and factors described above in planned (neuro-)aesthetic studies. This could be done by a tag list or flow diagrams addressing those variables. Such a pre-scriptive system will also assist us in detecting the limitations of any study to make more explicitly and specifically clear which ecological factors might have been ignored or should be interpreted with caution. In this realm, we should always balance experimental control with ecological validity, as sketched out in Figure 6.”. I have added a figure (Figure 6) illustrating how the right balancing between ecological validity and experimental control could look like. ***